



# Sea Level Rise in Europe: knowledge gaps identified through a participatory approach

Coordinating lead author: José A Jiménez[1]

Authors: Antonio Bonaduce[2], Michael Depuydt[3], Giulia Galluccio[4], Bart van den Hurk[5], H.E. Markus Meier[6], Nadia Pinardi[7], Lavinia G. Pomarico[8], Natalia Vazquez Riveiros[9], Gundula Winter[5]

[1]Laboratori d'Enginyeria Marítima, Universitat Politécnica de Catalunya·BarcelonaTech, Barcelona 08034, Spain.
[2] Nansen Environmental and Remote Sensing Center (NERSC), Bergen, Norway.
[3] Joint Programming Initiative – Climate, Brussels, Belgium.
[4] Euro-Mediterranean Center on Climate Change, 20123 Milano, Italy.
[5] Deltares, Boussinesqweg 1, Delft, The Netherlands.
[6] Leibniz Institute for Baltic Sea Research Warnemünde, 18119 Rostock, Germany.
[7] Decade Collaborative Center for Coastal Resilience, Department of Physics and Astronomy, University of Bologna, Italy.
[8] Joint Programming Initiative Healthy and Productive Seas and Oceans, Brussels, Belgium.
[9] UMR 6538 Geo-Ocean, CNRS, IFREMER, UBO, Plouzané, France.

*Correspondence to*: José A Jiménez (jose.jimenez@upc.edu)

**Abstract**

The Intergovernmental Panel on Climate Change (IPCC) plays a pivotal role in delivering information and knowledge on sea level rise (SLR), a global threat impacting coastlines worldwide. However, considerable disparities still persist in Europe in understanding and applying sea level science, evaluating its repercussions, and devising effective adaptation strategies. These are influenced by local factors such as diverse environments, socioeconomic conditions, policy contexts, and diversity in stakeholder involvement, producing in turn varying knowledge gaps and information needs across European sea basins. In this context, this paper presents the findings of a comprehensive scoping process carried out by the European Knowledge Hub on Sea Level Rise (KH-SLR) to define the outline of the first KH-SLR Assessment Report. It consists of the analysis of stakeholder responses to an online survey and insights shared during four regional workshops, aiming to pinpoint critical gaps in available information on SLR and its potential consequences in European basins, both from a scientific and policy perspective. The analysis was divided into three categories: i) SLR science and information, ii) SLR impacts, and iii) SLR adaptation policies and decision-making. Regarding SLR science and information, many respondents found that significant gaps exist in regional SLR projections and uncertainties, particularly related to long-term SLR induced by potential melting of large icesheets. Interestingly, the perspective on information gaps is different for scientists (emphasizing the need to increase regional projection capabilities) and government users (stressing the availability of accurate projections for their regions). Regarding impacts and hazards, shoreline erosion stands out as a dominant concern in all basins except the Arctic, while emerging issues like saltwater intrusion and the role of SLR in compound risks associated with extreme water levels and river flow were also given significant regional relevance. With regards to policy and decision making, existing adaptation plans are perceived as ineffective and lacking adaptability, with gaps related to underestimated impacts and urban planning. Participants, especially end-users, emphasized the relevance of improved information dissemination and communication to support informed decision-making.

## 1. Introduction

Despite the global threat posed by sea level rise (SLR) to coastlines worldwide and the crucial role played by the IPCC in providing assessments based on existing literature (IPCC, 2021; IPCC, 2022), there is an uneven coverage in the knowledge





and utilization of sea level science, the assessment of its impacts, and the development of adaptation plans (Magnan et al., 2023; McEvoy et al., 2021). This may be associated with local factors such as the diversity of environments, socioeconomic conditions, policy contexts, and stakeholders, which cause local needs and knowledge gaps to vary from one site to another.

As decisions regarding the response to SLR need to be made at a national, regional or local scale, it is necessary to assess knowledge gaps and needs at the same scale. This the ambition of the European Knowledge Hub on Sea Level Rise (KH-SLR) which was initiated with the objective of providing easily accessible and practical knowledge on regional and local sea level changes and their consequences. For each of the ocean and sea basins surrounding Europe (Fig. 1, Table 1), characteristics on drivers of sea level variability, coastal occupation, sea level rise impacts and approaches to sea level rise adaptation are

recognized.

To achieve its long-term goals (see Sea Level Rise in Europe: a Knowledge Hub at the ocean-climate nexus), the initial implementation phase of the KH-SLR centred on a scoping process. This process consisted of three key components that collectively contributed to identifying the primary issues pertinent to European Seas. The approach followed a bottom-up methodology, which integrated the viewpoints and contributions of representative stakeholders from European Seas. As

suggested by (Fraussen et al., 2020), an effective stakeholder consultation approach involves a hybrid array of tools, encompassing open surveys, workshops, conferences and closed consultations with specific interest groups. This comprehensive approach enhances engagement with a diverse range of stakeholders and ensures a rich inflow of information.

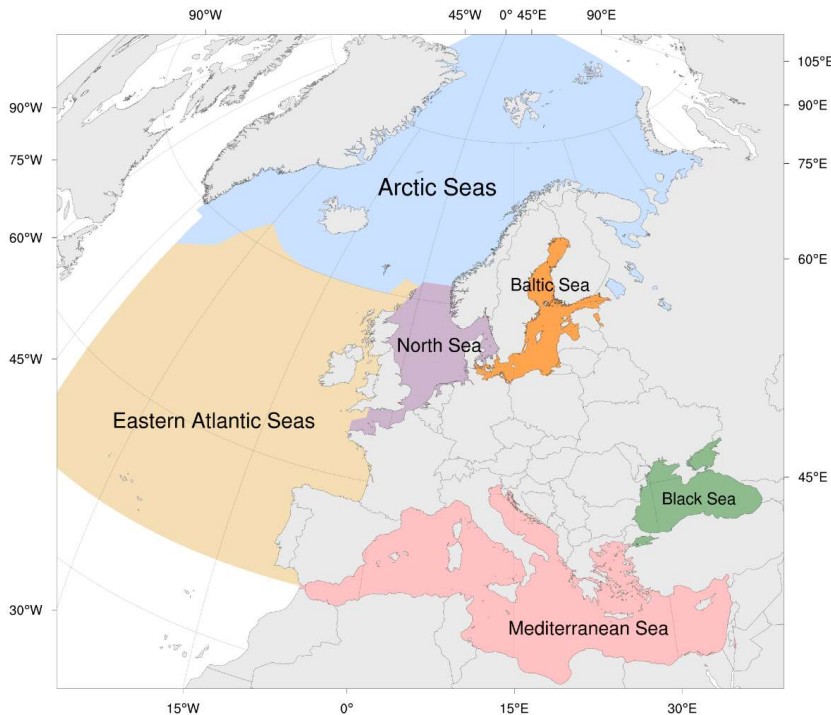

**Figure 1. Europe regional seas**



The KH-SLR scoping process adopted this hybrid approach through four key components: (i) an online survey, designed to
collect insights and perceptions on SLR in European basins from a diverse range of stakeholders; (ii) four dedicated workshops
on SLR, tailored to each basin, which provided focused discussions and knowledge exchange, enabling a deeper understanding
of regional challenges; (iii) a pan-European conference on SLR, serving as a platform for experts and stakeholders from across
Europe to share their expertise, experiences, and perspectives on SLR; (IV) a closed consultation with Member State
representatives involved in the Joint Programming Initiatives (JPI), JPI Climate and JPI Oceans.

This work provides a comprehensive summary of the scoping process undertaken in the survey and basin-specific workshops,
and presents the key findings from each. The primary objective of this process is to identify critical gaps in available
information on regional SLR and its potential impacts in European basins and to discern the knowledge requirements and areas
necessitating further research for both experts and stakeholders. These findings form the basis for this assessment report and
are expected to inform future research endeavors and policy decisions.


**Table 1. Basic indicators per European sea basins (data sources and methodology are shown in Supplementary material) (LECZ: low elevation coastal zone, between 0 and +10 m above MSL; GIA: glacial isostatic adjustment). Rates of SLR per European regional seas for 19502-2014 based on (Dangendorf et al., 2019). Coastal archetypes as defined in Haasnoot et al (2019). Methods to derive extension of archetypes and population are shown in the Supplementary material. (\*) The extension of the coastal zone along the**
**European Sea basins used to measure archetypes and population in LECZ is shown in figure S2 (supplementary material).**

| Basin name and countries (*) | Mean SLR 1950-2014 (mm/year) | Coastal archetypes (%) | Population in LECZ (2020) |
|---|---|---|---|
| North Sea (*Denmark, UK, Germany, Norway, Netherlands, Belgium*) | 1.5 ± 0.1 | Urban: 6.44 %<br>Rural: 62.62 %<br>Urban delta: 0.49 %<br>Rural delta: 2.73 %<br>Urban estuary: 0.72 %<br>Rural estuary: 23.91 %<br>Urban delta/estuary: 0.41%<br>Rural delta/estuary: 1.82%<br>Cliff: 0.87 % | 24.88 M people |
| Arctic Seas (*Norway, Iceland*) | 1.5 ± 0.1<br>1.4 ± 0.1 (GIA corrected) | Urban: 4.29 %<br>Rural: 84.39 %<br>Urban estuary: 0.44 %<br>Rural estuary: 5.90 %<br>Cliff: 4.97 % | 9.02 M people |
| Atlantic coast (*France, Spain, Ireland, UK, Portugal*) | 1.2 ± 0.1 | | |
| Baltic Sea (*Sweden, Denmark, Finland, Latvia, Estonia, Lithuania, Poland, Germany*) | -1.1 ± 0.4<br>1.8 ± 0.4 (GIA corrected) | Urban: 6.26 %<br>Rural: 77.09 %<br>Urban delta: 0.11 %<br>Rural delta: 0.66 %<br>Urban estuary: 1.03 %<br>Rural estuary: 14.19 %<br>Urban delta/estuary: 0.01%<br>Rural delta/estuary: 0.46 %<br>Cliff: 0.18 % | 6.90 M people |
| Mediterranean Sea (*Spain, France, Italy, Croatia, Montenegro, Albania, Greece, Malta, Turkey*) | 1.2 ± 0.1 | Urban: 6.55 %<br>Rural: 73.95 %<br>Urban delta: 0.07 %<br>Rural delta: 1.00 %<br>Urban estuary: 0.38 %<br>Rural estuary: 17.31 %<br>Urban delta/estuary: 0.05%<br>Rural delta/estuary: 0.60 %<br>Cliff: 0.54 % | 12.38 M people |
| Black Sea (*Romania, Bulgaria, Turkey*) | 1.2 ± 0.1 | Urban: 7.45 %<br>Rural: 78.57 %<br>Urban delta: 0.03 %<br>Rural delta: 2.11 %<br>Urban estuary: 0.90 %<br>Rural estuary: 1.55 %<br>Urban delta/estuary: 0.02%<br>Rural delta/estuary: 9.34 %<br>Cliff: 0.05 % | 1.31 M people |



**2. Methods**

**2.1 Survey design and data collection**

The KH-SLR conducted an online survey targeting stakeholders involved in coastal planning and research, especially those whose work is related to or influenced by SLR. The online questionnaire was hosted on the EU Survey platform
(https://ec.europa.eu/eusurvey/runner/KH-SLRsurvey2022). Invitations to participate were distributed through various channels, including the JPI Climate and JPI Oceans websites and social media channels, direct outreach to individuals within government offices, and distribution via mailing lists. Invited participants were also encouraged to share the survey with others who identified within the target audience. The first round of invitations was dispatched in January 2022, followed by multiple reminders in the first half of 2022. The data presented here reflect responses received until July 2022, in anticipation of the
Sea Level Rise Conference 2022 held by the KH-SLR in October 2022 in Venice, Italy.

The survey questionnaire commenced with a concise introduction, outlining its purpose. It was structured in four sections. The first section sought information about the respondents, including the type of institution/organization they were affiliated with and the specific sea basin that best aligned with their work. For both questions, participants had the option to select multiple responses when applicable. The second section consisted of five closed-ended questions and one open-ended question, with
the aim of assessing the need for, availability of, requirements for, and usage of SLR information. The third section featured three closed-ended questions, serving the purpose of identifying the most significant impacts associated with SLR. It also assessed the availability and importance of impact assessments. The final section included three closed-ended questions and two open-ended questions, focused on policy decisions and adaptation strategies related to SLR. The survey concluded with a general question about the perceived usefulness of SLR information on IPCC Assessment reports. A comprehensive list of all
survey questions can be found in the supplementary material.

In total, we received responses from 200 participants across 23 European countries (94 % of the participants) and 8 non-European countries (6% of participants), who provided information and perceptions about the covered basins according to the distribution shown in Figure 2. The participants were broadly categorized in two professional groups (Figure 2): (i) government, encompassing individuals working within regional or central government agencies and international
organizations (about 35% of the total); and (ii) research, including those affiliated with universities, research institutes, private companies and NGOs (about 64 % of the total).

To assess closed-ended questions related to specific topical statements, a Likert-type scale with five response categories was employed, spanning from "strongly disagree (1)" to "strongly agree (5)." Likewise, a similar scale was utilized to gauge the perceived significance of the impact assessment, offering choices from "not important (1)" to "very important (5)." Similarly,
when evaluating the effectiveness of adaptation strategies, the scale ranged from "nonexistent (1)" to "very effective (5)."

To determine the overall relevance of multiple answers, a *total score* was calculated, that considered responses from all surveyed basins. This score was computed by summing the percentages of respondents who selected each answer across all basins. The resulting score ranges from 0 (indicating that no respondents chose the answer in any basin) to 600 (indicating that 100% of respondents across all basins selected that answer).

Regarding open-ended questions, we categorized responses by keywords that encapsulated their content. These keywords were then visualized using a word cloud chart to highlight the most pertinent topics, while estimating the percentage of times they were identified by participants.



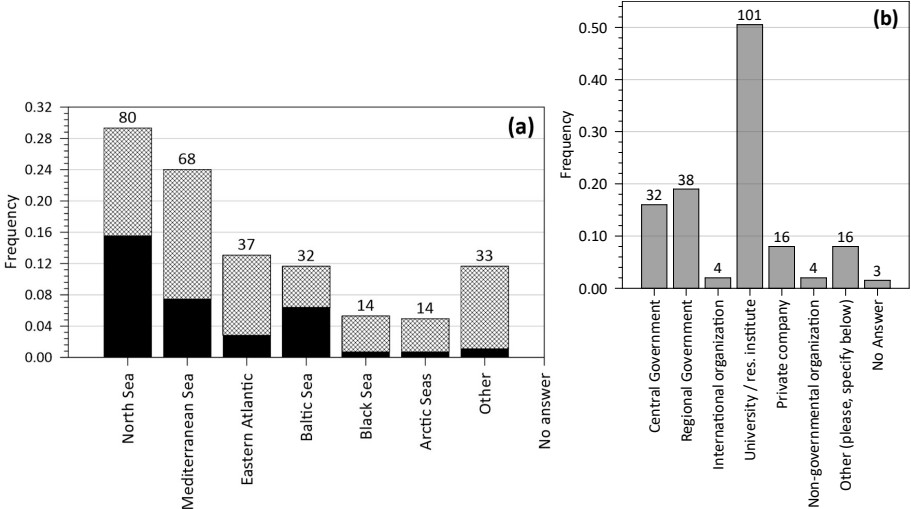

**Figure 2. (a) Breakdown of respondents by sea basin (solid color bars: % government respondents; cross-hatched bar: % research respondents). (b) Distribution of respondents by organization type. The numbers above each bar indicate the total number of respondents for each category (sea basin and organization type). Note that respondents can be representative of more than one basin and may belong to two different institutions.**

## 2.2 Scoping workshops

The scoping workshops conducted in 2022 played a pivotal role in the process of identifying the requirements of policy makers, coastal planners, and stakeholders at large. The insights gathered from these workshops were instrumental in shaping and collaboratively designing the key themes related to sea level rise drivers, impacts, and policy options for each of the Europe's major ocean basins to be addressed in the Assessment Report.

Four scoping workshops were run online between March and May 2022. Each workshop had a specific focus on one or two European sea basins and was organized by one or more partner institutes within the respective region, with support from the Secretariat to the Knowledge Hub on Sea Level Rise (table 2).

**Table 2. List of scoping workshops.**

| Region | Organizers | Dates | N registered attendees |
|---|---|---|---|
| North Sea & Arctic Ocean | Deltares, NL and Nansen Environmental and Remote Sensing Center, NO | 21-22 March 2022 | 65 |
| Eastern Atlantic | French Research Institute for Exploitation of the Sea, FR | 28-29 April 2022 | 42 |
| Mediterranean & Black Seas | Universitat Politècnica de Catalunya·BarcelonaTech, ES; University of Bologna, IT, and Euro-Mediterranean Centre on Climate Change, IT. | 5-6 May 2022 | 70 |
| Baltic Sea | Leibniz Institute for Baltic Sea Research Warnemünde, DE; Federal Maritime and Hydrographic Agency of Germany, DE, and Tallinn University of Technology, EE. | 9-10 May 2022 | 70 |



The agenda of the workshops mirrored the structure of the survey, although each specific workshop adapted it slightly. This approach ensured that results would be comparable and allowed for a cohesive discussion of the three main sections: i) SLR physical science and data; ii) SLR hazards and impacts; iii) SLR adaptation policies and decision-making. The agenda was

further divided into distinct segments, including keynote speeches, stakeholder contributions, and expert presentations from the scientific community. In addition to these, interactive breakout sessions were incorporated, moderated by the workshop conveners. These interactive sessions were facilitated using the remote collaboration tool Mural. The detailed agendas of scoping workshops can be seen in the Supplementary material.

Each online workshop spanned two days, totalling eight hours of engagement, and attracted a diverse range of participants,

with attendance ranging from 42 to 70 registered individuals (table 2). Stakeholders from each European basin, including respondents from the survey, received personalized invitations via email to register for the workshops. Upon approval of their registration, participants received comprehensive materials, including the agenda, meeting link, detailed instructions, and expectations from their active involvement in the workshop.

### 3. Results

**3.1 Survey**

SLR information

When asked about the availability of essential information and data on SLR required for their work, approximately 32% of the respondents expressed that a substantial portion of this information is missing. This observation holds true across different respondent profiles (government: 33%; research: 32%) (see Table S2 in supplementary material). The highest percentage

reporting a lack of information was identified in the Arctic (43%) and Mediterranean (40%) sea basins. Notably, there was a significant difference between science (34%) and government (57%) respondents in these regions, emphasizing the disparity in access to information. In contrast, the lowest percentages of respondents indicating information deficits were associated with the Baltic Sea (25%) and North Sea (26%) basins (Fig. 3).

Among the various types of available information, global sea level projections received the highest accessibility and utilization

scores (total score of 455/600). Regional sea level projections followed closely (total score of 367/600) as depicted in Figure 3. Importantly, there were no significant disparities observed across different sea basins with differences remaining under 15%. However, it's worth noting that the Black Sea and Arctic basins exhibited the largest deviations from the prevailing trend regarding information accessibility (global and regional projections as information types). Nevertheless, these findings show the disparity in the use of SLR information among stakeholders across different basins. (Hirschfeld et al., 2023) previously

pointed to this inconsistency in the use of SLR information by coastal planners in their adaptation efforts.



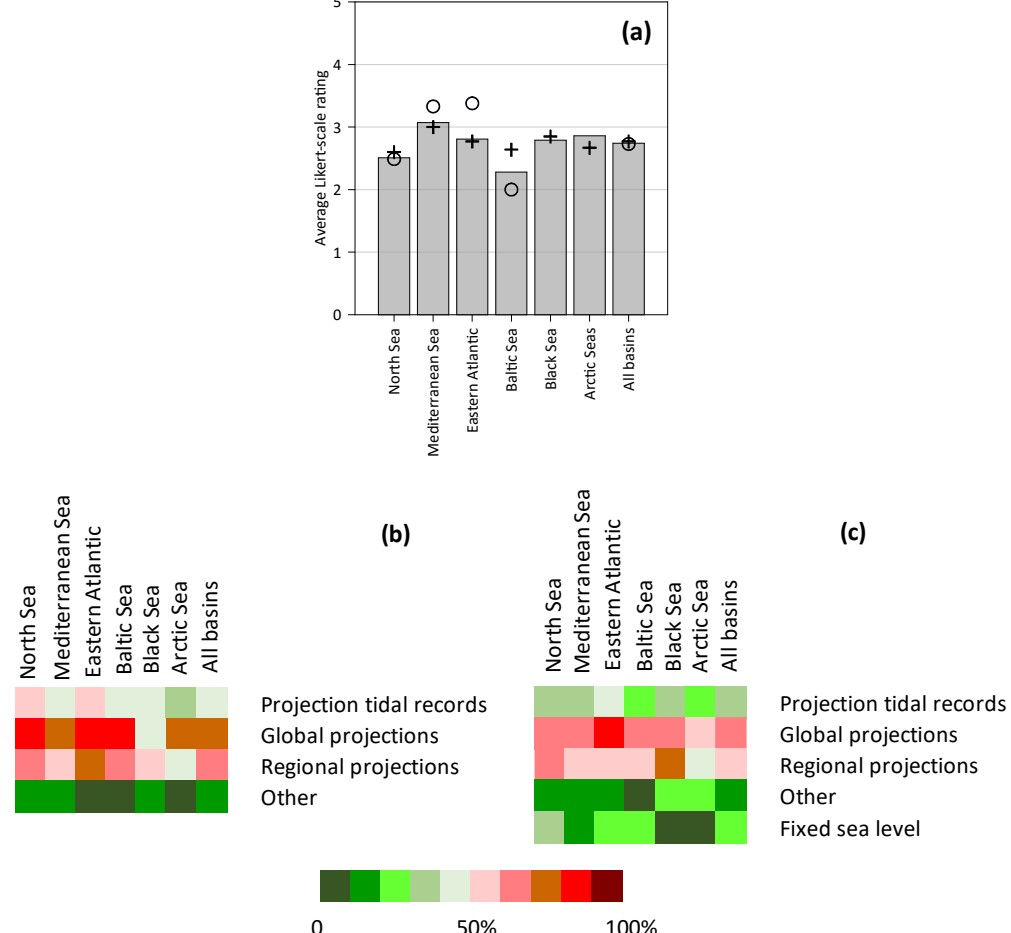

**Figure 3. (a) Average rating on the Likert-scale to the statement "For my work crucial information and data on SLR is missing and/or not accessible" (o: government; +: scientists; grey bar: total). (b) Percentage of respondents who reported having access to specific types of SLR data/information (original question was "What type of SLR data and/or information do you have access to?") (c) Percentage of respondents who reported the use of the mentioned type of SLR data/information (original question was "What type of SLR data and/or information do you use?")**


All respondents unanimously concurred on the necessity for periodic updates to SLR projections and the importance of comprehending the associated uncertainties in these projections (see Table S2 in supplementary material). Over the years, SLR projections and their uncertainty have undergone notable evolution, as evidenced by (Garner et al., 2018) and (Bamber et al., 2022) among others, emphasizing the need for regular updates.

Figure 4 shows the word clouds generated from responses to an open-ended question seeking to identify the most significant knowledge gaps in SLR among respondents from both science and government. The percentage of responses identifying each keyword-related issue per respondents' category is shown in Table 3. The identified gaps are notable in three topics: regional and local SLR projections, the overall level of uncertainty associated with these projections, and, most significantly, the uncertainty related to contributions from ice sheet melting. Both government and scientist respondents identified the same

gaps, albeit with slight variations in their perspectives and relative importance (Table 3). For instance, government respondents



emphasized the need for precise regional projections, viewing them as the ultimate product. In their perspective, these projections play a crucial role in fulfilling their responsibilities and, in relation to this, uncertainty emerges as the second most identified issue and, in this case, these stakeholders are concerned about how to address it. On the other hand, scientists prioritize a more comprehensive understanding of the various factors influencing regional projections, considering these
insights as the final goal to be achieved. Uncertainty is highly mentioned, especially with regard to the factors contributing to it. In addition to these commonly recognized gaps, scientists expressed heightened concern regarding other common issues. These include improving local SLR projections, which requires a more accurate understanding of ground level movements. Surprisingly, government respondents appear to be less concerned about this matter. Furthermore, both types of respondents acknowledge the necessity of comprehending the impact of SLR on extreme water levels, as well as its influence on
compound/cascading events, and multi-hazard risks, although the latter is given lower priority.

**Figure 4. Word cloud representation of responses to the open-ended question "From the perspective of your work, what are the largest knowledge gaps in SLR?" from scientists (left) and government (right) respondents (Generated using the WordArt Generator by WordArt.com) (see table 3 for their quantitative representativity).**

**Table 3. List of keywords and percentage of responses within the type of respondents who identify a keyword-related issue to the open-ended question "From the perspective of your work, what are the largest knowledge gaps in SLR?" (only issues with a response rate larger than 5% are shown). Examples of different responses associated to the same keyword indicating a different view/interest in the issue.**

| Respondents' profile | Scientists | | Government | |
|---|---|---|---|---|
| Keywords and % of responses identifying a keyword-related issue over the total of responses | Regional projections<br>Local projections<br>Ice sheet contribution<br>Extreme sea levels<br>Uncertainty<br>Impacts<br>Ground motion | 19 %<br>13 %<br>11 %<br>10 %<br>10 %<br>7 %<br>6 % | Regional projections<br>Uncertainty<br>SLR acceleration<br>Extreme sea levels<br>Ice sheet contribution<br>Impacts<br>Longer-term projections | 29 %<br>18 %<br>11 %<br>9 %<br>9 %<br>9 %<br>5 % |
| Example of different views on the same topic: *regional projections* | Determining relative importance of different regional contributions (land subsidence, isostatic adjustment, glacier melting, sediment compaction). | | Regional mean sea level projections for the inner German Bight for different IPCC scenarios. | |
| Example of different views on the same topic: *uncertainty* | Refining uncertainty in future sea level projections associated with deep ocean contribution, Arctic contribution, ice sheet mass change. | | The largest gap is not the question of understanding how uncertain any given SLR scenario is, but rather dealing with the fact that all SLR scenarios are uncertain. | |





Impacts

The experts assessed the most significant impacts of SLR for each of the basins by selecting from a list of the most common impacts along the European coast (Fig. 5). Among these impacts, *coastal/beach erosion* emerged as the most critical concern,

with a total score of 537/600, prevailing in all basins except the Arctic Sea. The prominence of this issue can be attributed to the essential role played by beaches, not only in supporting coastal tourism and the regional economy but also in providing a natural defence for inland areas. Furthermore, this is a widely recognized SLR-induced impact (e.g. (Nicholls & Cazenave, 2010), the importance of which has been documented along the European coastline (e.g. (Vousdoukas et al., 2020a). The reduced significance of this impact in the Arctic Seas can be attributed to the fact that this region has the lowest percentage of

sandy shoreline (e.g. Luijendijk et al., 2018).

The second most pertinent impact identified was the influence of SLR in *increasing storm impacts*, a concern uniformly acknowledged across all basins (total score of 480/600). This impact is well-documented and widely acknowledged, involving the projected rise in extreme water levels due to SLR, thereby increasing the likelihood of present-day storm surges and inundation events (e.g. Vousdoukas et al., 2018). Conversely, *permanent inundation* due to SLR is generally perceived as a

less significant impact (361/600). It will primarily affect very low-lying and unprotected areas, with relatively limited extent, mainly concentrated in natural areas (e.g. Antonioli et al., 2020).

*Damage or loss to public infrastructure* (471/600) and, in a slightly smaller proportion, *private properties* (417/600), were identified as relevant impacts. This is highly related to the large exposure of these assets along the European coasts and with expected increase in damage under SLR (e.g. Vousdoukas et al., 2020b)

*Groundwater salinisation* (338/600) is a lesser concern in the Eastern Atlantic, Black Sea and Arctic basins. In contrast, it holds substantial importance in the remaining sea basins. This significance is grounded in the presence of pre-existing soil salinisation issues (Daliakopoulos et al., 2016), and the anticipation of potential salinity challenges exacerbated by climate-related factors (e.g. Falloon & Betts, 2010; Oude Essink et al., 2010). The relatively limited attention given to this impact can be linked to the predominant role played by other natural and anthropogenic variables that affect groundwater salinity (e.g.

Taylor et al., 2013).

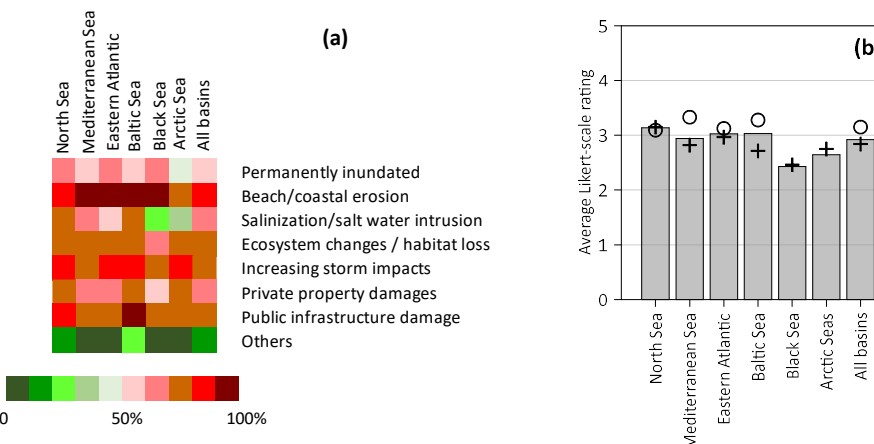

**Figure 5. (a) Relevance of SLR-induced impacts in each sea basin indicated by the percentage of respondents who identified these impacts. (b) Average rating on the Likert-scale to the statement "High quality and up-to-date assessments of SLR-induced impacts are available for making decisions on planning"). (o: government; +: scientists; colour bar: total)**





The significance of these impacts for the European sea basins is underscored by the nearly unanimous consensus among respondents (mean value of 4.55 in the Likert scale) on the need to employ impact assessments in shaping planning decisions amidst SLR (see Table S3 in supplementary material). Despite this consensus, approximately 39% of all respondents faced challenges due to the absence of current and high-quality and up-to-date assessments of SLR-induced impacts. This perception was consistent across all sea basins, with the Black Sea and Arctic Sea facing the most pronounced gaps in available

assessments (Fig. 5). Government respondents expressed more optimism compared to those from the research sector (see Table S3 in supplementary material). Specifically, 44% of research respondents disagreed or strongly disagreed with the statement "existing high-quality and up-to-date assessments of SLR-induced impacts," whereas only 32% of government respondents held this view.

Adaptation

Lastly, respondents were queried on the performance of adaptation plans and strategies aimed at addressing the impacts of SLR in their respective regions (see Table S4 in supplementary material). Regarding the *effectiveness* of current adaptation plans, a noteworthy 51% of respondents assessed them as either insufficient or inexistent (Fig 6). Significantly, scientists exhibited a more critical perception in this regard, with an additional 18% deeming the plans as insufficient compared to government respondents. Nevertheless, a relatively low proportion of respondents (7.5%) indicated the complete absence of

adaptation plans, aligning with the findings of a recent survey of McEvoy et al. (2021) on the planning approaches of European countries in response to SLR. Notably, the Black Sea basin emerged as the region where the absence of plans was most conspicuous.

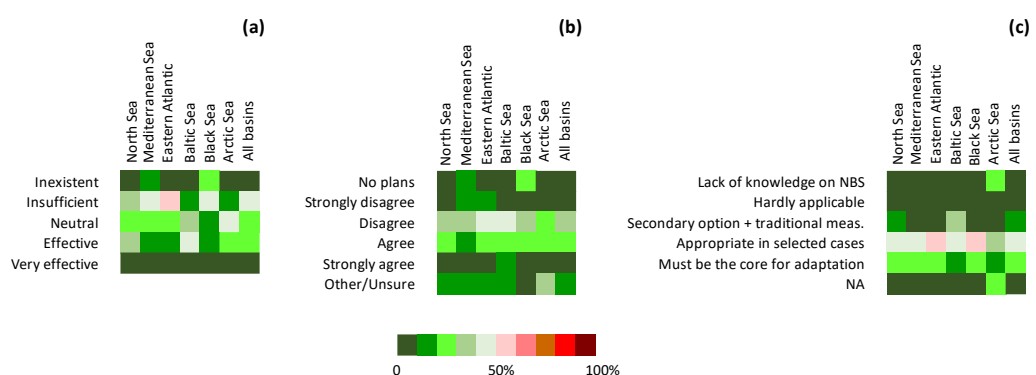

**Figure 6. Percentage of responses by sea basin for the following questions/statements: (a) "How effective do you consider the present**
**adaptation strategy to SLR in your country/region?". (b) "Existing adaptation strategies/plans are flexible enough to adapt to future updates in SLR-induced impacts, or to cope with the inherent uncertainty in their assessment". (c) "Nature-based solutions (NBS) are appropriate as adaptation measures to SLR in your country/region".**

Regarding the perceived flexibility of existing adaptation strategies and plans in the face of future SLR-induced impacts (or
vice versa, the ability to cope with the inherent uncertainty in their assessment), 40% of respondents expressed the view that existing plans lack sufficient flexibility (see Table S4 in supplementary material). This perception remained relatively consistent across different sea basins, with the Arctic and Black Sea exhibiting the lowest perceived lack of flexibility. In general, there were no significant differences in perception based on respondent type, except in the North Sea where government respondents were notably less optimistic about flexibility, with a 15% difference compared to scientists. It is

important to note that flexible adaptation allows for plan adjustments in response to future changes. Unless plans are designed





with an adaptation pathways-like approach (Haasnoot et al., 2013), achieving this flexibility can be challenging. In this context, Kim et al. (2022) introduced a framework for assessing flexibility in adaptation plans.

Participants were asked to identify areas where considerations related to SLR are often neglected but should be incorporated into decisions and policy objectives. Fig 7 shows word clouds generated from responses to an open-ended question, while

Table 4 provides the percentage distribution of responses according to the type of respondents. A significant proportion of respondents (68 % and 65 % for scientist and government respectively) either did not respond to this question or indicated that there were no relevant decision requiring the inclusion of SLR considerations that did not include it. Notably, scientists identified a greater number of issues in comparison to government respondents (Fig 7). Those who identified such omissions emphasized key gaps primarily related to management issues in the coastal zone or, directly, SLR-induced impacts such as

salt-water intrusion or damage to infrastructure (previously prioritized in Fig 5). A prominent emerging issue is the interaction of SLR with coastal ecosystems, which is mentioned in different ways, including its impact on existing ecosystems, disruptions of ecosystem services, and ecosystem management. This aligns with the growing concerns about the anticipated impact of SLR on coastal habitats, particularly in areas such as coastal wetlands (e.g. Schuerch et al., 2018), and the projected decline in services provided by coastal ecosystems (e.g. Paprotny et al., 2021). Furthermore, urban planning is a notable concern, in line

with the expected impacts of SLR on coastal cities (e.g. Abadie et al., 2019). This indicates that the legal competence of cities in managing coastal issues is often insufficient and underlines the necessity of integrating SLR considerations in urban planning frameworks. Other identified concerns include the influence of SLR on river flow and flood management, a topic gaining increased attention in the context of compound risks (e.g. Bermúdez et al., 2021); and the effects of SLR on seawater intrusion and, consequently, in freshwater management (e.g. Ketabchi et al., 2016) and agriculture (e.g. Gopalakrishnan et al.,

285 2019).

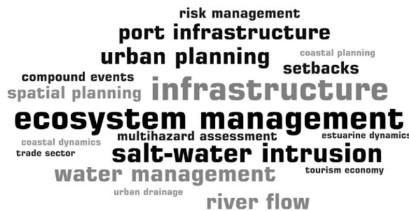

**Figure 7. Word cloud representation of responses to the open-ended question "Are there other decisions/purposes for which you currently don't consider SLR, but for which you think it would be important to do so?" from scientists (left) and government (right) respondents (Generated using the WordArt Generator by WordArt.com) (see table 4 for their quantitative representativity).**


**Table 4. List of keywords and percentage of responses within the type of respondents who identify a keyword-related issue to the open-ended question "Are there other decisions/purposes for which you currently don't consider SLR, but for which you think it would be important to do so?" (only issues with a response rate larger than 5% are shown).**

| Respondents' profile | Scientists | | Government | |
|---|---|---|---|---|
| Keywords and % of responses identifying a keyword-related issue over the total of responses | Infrastructures | 13 % | Ecosystem management | 23 % |
| | Ecosystem management | 13 % | Urban planning | 19 % |
| | Salt-water intrusion | 9 % | Infrastructures | 12 % |
| | Water management | 7 % | Agriculture | 12 % |
| | Urban planning | 7 % | Spatial planning | 8 % |
| | River flow | 7 % | Salt-water intrusion | 8 % |
| | Port infrastructure | 7 % | | |



Lastly, in response to the increasing recognition of nature-based solutions (NBS) (e.g. European Environment Agency, 2021), we included a specific question about their suitability as adaptation measure to address SLR-induced impacts. While all respondents recognized the value of incorporating NBS in coastal adaptation plans, the majority viewed their effectiveness as conditional, dependent on site-specific circumstances (Fig. 6c) (see Table S4 in supplementary material). This perspective emphasizes the importance of providing a more comprehensive account of the co-benefits and lessons learned from prior

implementations of NBS measures (e.g. Moraes et al., 2022). Furthermore, it calls for a rigorous evaluation of their effectiveness when compared to artificial protection structures (e.g. Morris et al., 2018) and substantiated evidence of their long-term cost-effectiveness and self-sustainability (e.g. Toimil et al., 2020).

Finally, it is worth noting that all respondents quasi-unanimously acknowledged the high level of usefulness of IPCC reports for their work, as evidenced by an average rating of 4.4 on the Likert scale (see Table S4 in supplementary material). This

consensus is consistent across different basins and respondent types.

**3.2 Workshops**

In this section, key points derived from the workshop discussions are presented. While the discussions were extensive and covered a wide range of issues, we focus on points that complement the survey results presented in the previous section or are

considered relevant for further specification. Results are presented following the three main themes: SLR information, hazards & impacts; and adaptation.

**North Sea and Arctic basins**

SLR information

Recurrent themes in the sessions focusing on the physical science of SLR included the need for locally specific reconstructions

and projections of extreme sea levels. It was recommended to incorporate local observations when studying historical extreme events, as this forms the foundation for precise impact assessments and statistical analysis. Additionally, research-oriented attendees expressed their desire for comprehensive guidance regarding existing models, recent developments, their limitations, and how to interpret model outputs. This is particularly crucial when dealing with low probability, high-impact scenarios and sea-level milestones.

Hazards & impacts

With respect to hazards and impacts, regional assessments should encompass a comprehensive understanding of the interplay of various processes that contribute to the magnitude of sea-level extremes. This includes accounting for vertical land movements, shifts in wind patterns, and the spatial extent of compound flooding events in coastal areas. While its' true that the consequences of SLR, such as erosion, salt intrusion, and flooding, may differ among regions, there is potential for mutual

learning and information exchange. This includes sharing data, tools, and the development of a European catalogue of significant historical events.

Adaptation and decision making

During the sessions focused on policy and adaptation, a clear consensus emerged regarding the need for a comprehensive overview of adaptation options. Such an overview should encompass details on the suitability of individual options in specific

environments, the scalability of pilot initiatives, an evaluation of the co-benefits and drawbacks associated with each measure, and real-world examples of successful applications. Policy-makers demonstrated a particular interest in exploring NBS and





sought guidance on structuring the adaptation planning process, for example, through Dynamic Adaptive Policy Pathways (Haasnoot et al., 2020). The participants also expressed a desire for a comparative assessment of policies across different countries to facilitate shared learning and to evaluate and compare the progress in adaptation across countries. To encourage

community and stakeholder engagement, attendees stressed the importance of transparent communication and the use of clear visualizations. Policy-makers specifically emphasized the need for geo-visualization tools that support decision-making and communication. They also requested scientists to provide clear explanations of how global sea level rise data is downscaled and how this data is interpreted within a local context.

**Eastern Atlantic basin**

SLR information

The discussions highlighted several knowledge gaps that have significant implications for future SLR management. These gaps encompass the need for comprehensive SLR scenarios tailored to estuaries, as well as the necessity of conducting local-scale assessments to bridge geographic information disparities. Furthermore, the discussions underscored the importance of enhancing the spatial resolution of climate models and projections, as well as incorporating low likelihood scenarios. The

monitoring of ice sheets and other key processes was actively discussed in the context of the set-up of early warning systems. Key areas for advancement were identified, including the imperative to improve ice sheet modelling, to gain a deeper understanding of climate system tipping points, especially in the context of ice sheets, and to update sea level budget (e.g. WCRP Global Sea Level Budget Group, 2018) along coastlines.

Hazards and impacts

A strong consensus emerged regarding the pivotal need to better assess the combined impact of waves, surges, tides and mean sea level rise. Ideally, future planning should consider the potential for internal variability in compound flood hazards, such as the combination of storm surges with river discharges and SLR, including changing trends in storminess. Cascading impacts involving SLR and human activities, such as salt intrusion affecting agriculture, was widely acknowledged but often overlooked in planning. The protection of cultural heritage requires specific actions, yet the implementation of informed

preservation strategies seems to face obstacles due to the absence of systematic and localized assessments.

Adaptation and decision making

Throughout this session, it became evident that the adequate identification and improved engagement of stakeholders are fundamental prerequisites for the adaptation process, requiring additional efforts. Participants stressed the importance of enhancing the language used in communication, particularly when reaching out to the general public and policymakers.

National debates on SLR adaptation were also deemed crucial. A key focus was on clearly presenting the co-benefits of adaptation and delineating the costs of taking action and, just as crucially, the cost of inaction. The need to increase confidence in SLR projections was also highlighted. Related to this, there was a unanimous consensus on the necessity of developing multiple SLR scenarios tailored to different stakeholder groups. Governmental agencies, already actively involved in political measures against sea level impacts, require a different level of information than local communities, who may not fully grasp

the urgency of SLR due to perceiving it as similar to present-day floods.

**Mediterranean and Black Sea basins**

SLR information

The gaps and needs raised by stakeholders during the sessions related to SLR information can be grouped in four main categories. An integrative *data management* approach was recommended to facilitate the integration of different data types, to



establish standards for defining metadata and quality control, and endorsing a data policy promoting the free and open exchange
       of sea level data at the European level. Regarding *sea level data gaps*, key objectives should focus on sustaining the current
       tidal stations network (see Pérez Gómez et al., 2022), improving data distribution, and expanding spatial coverage, especially
       along the northern African coast. This includes the establishment of "open sea" tidal stations to enhance large-scale sea level
       monitoring. Standardized quality control procedures and data processing methods are essential (e.g. IOC, 2020).

There is a need for robust local *sea level projections* with quantified uncertainties, as well as examining low-probability, high-
       impact scenarios, and comprehensive numerical modelling of extreme water levels that considers various contributing factors
       like meteo-tsunamis, and river discharge – sea level interaction. Digital Twins could be considered for testing coastal
       adaptation options (e.g. Pillai et al., 2022). To comprehensively address SLR impacts and risks, there is a need for
       *multidisciplinary data* and model simulations. While the European Marine Observation and Data Network (EMODnet)

provides human activity data, their potential for SLR risk assessment remains untapped. Coastal vulnerability data is scarce
       and lacks standardization. Considering factors like sediment balance is crucial for long-term coastal erosion estimates, yet
       accurate data on sediment balance is often lacking. It is strongly recommended to establish requirements for high-resolution
       bathymetry and digital terrain models tailored for SLR and inundation analysis.

       Hazards & impacts

In relation to SLR impacts, attendees confirmed impacts identified in the survey, specifically erosion and flooding. Erosion
       was recognized as a critical factor that diminishes the coastal resilience to SLR and heightens its vulnerability. Additionally,
       discussions highlighted the significance of compound flooding, especially taking into account its occurrence along the basin.
       Participants also underscored the importance of addressing the impact of salt-water intrusion on freshwater resources due to
       SLR, especially in light of the expected increase in desertification in these basins (e.g. Gao & Giorgi, 2008). In the context of

assessing risks and impacts, it was deemed essential to consider *'what if' scenarios* for SLR, including extreme SLR scenarios.
       Given the prevalence of low-lying sedimentary features like deltas and coastal plains in the region, controlling and measuring
       local *vertical land movements* was considered crucial. Also, an accurate estimation of the *vulnerability of the densely populated*
       coastal zones, and their *exposure* and values were also considered as a top priority.

       The second part of the session was dedicated to eliciting crucial information required for assessing hazards, risks, and impacts.

Notably, inputs often mirrored the participants' local experiences, emphasizing the significance of accessing specific data that
       might already be available and accessible in other locations. This highlights a key characteristic of the region: stakeholders
       from various countries and institutions exhibit a diverse spectrum of profiles in terms of data accessibility, assessment
       methodologies, and their commitment to conducting assessments at different scales. Significant knowledge gaps related to
       hazards and vulnerability were particularly evident in the southern Mediterranean Sea and non-European Sea coastal areas.

Adaptation and decision making

       Several key themes emerged as significant areas requiring attention in the forthcoming assessment report with regard to SLR
       adaptation strategies and policies. Foremost among these was the imperative of incorporating the needs and challenges of
       future generations into the frameworks. The second priority highlighted the necessity to bridge the knowledge gap by
       standardizing the information derived from observations and models, with the aim of informing and prioritizing action.

Integrated Coastal Zone Management was underlined as a foundational paradigm for the development of new policy
       instruments aimed at bolstering coastal resilience, as an integral component of Marine Spatial Planning strategies. Additionally,
       any adaptation policy should take into account social factors and community engagement, ensuring a participatory decision-
       making process where diverse stakeholders have a voice. This approach also requires the implementation of effective outreach
       and communication strategies.





**Baltic Sea basin**

SLR information

Participants highlighted that there is a need to constrain the uncertainty in sea level rise (SLR) along the Baltic coast, primarily arising from various sources, including the relative contributions of melting from the Greenland and Antarctic ice sheets and regional differences in the response of sea levels to atmospheric forcing, among others (e.g. Weisse et al., 2021). It was

considered necessary to have high resolution projections of future total water level extremes including wind contribution to properly reflect the spatial variability of sea level variations across the basin. The need to separate the effects of natural variability and anthropogenic global warming on long-term sea level changes was also emphasized. In addition, participants highlighted the need for progress in the characterization of drivers involving sea level variations triggering natural hazards, which might be amplified under SLR, including meteo-tsunamis and storm surges.

Hazards & impacts

In addition to well-documented erosion and flooding risks along the Baltic coast, other often overlooked impacts of SLR, such as salt-water intrusion and freshwater salinization, will be equally important for some areas. Compound events, such as the combined effects of extreme sea levels and high river discharges, pose a threat to coastal communities like Stockholm, Pärnu and Klaipeda, among others, especially in scenarios of rising sea levels and increased precipitation.

In the Baltic Sea, key locations such as St Petersburg, Stockholm and the Kiel Canal have already experienced or are projected to face substantial impacts from extreme sea levels and SLR. A recurring theme across these locations is the utilization of locks and water control infrastructure as a means to mitigate and adjust to elevated water levels. These critical infrastructures play a vital role in safeguarding coastal cities, preventing saltwater intrusion, and regulating levels for shipping across the region. Consequently, the challenge lies in effectively adapting to SLR while preserving the functionality of these vital systems.

Adaptation and decision making

Several topics related to adaptation were raised, often applicable to any basin. Enhancing the response to SLR involves integrating SLR-related policy and marine spatial planning, traditionally more focused on marine ecosystems. Identifying and addressing conflicts of interest, such as conservation versus economic development, is essential. Identifying the obstacles hindering implementation and devising workable solutions can help ensure the success of these initiatives.

Striking a balance between communicating scientific uncertainty and providing specific policy-compliant data is challenging but crucial. Overemphasis on uncertainty can potentially hinder adaptation efforts. It is recommended combining short-term and long-term planning, with a focus on adaptive planning approaches. Assessing the outcomes of SLR-related adaptation measures and policies, particularly for innovative measures like nature-based approaches, is critical. This includes an examination of their scalability and applicability across different contexts.

Recognizing the role of insurance and banking sectors in SLR policy and planning is pivotal for future coastal development. Effective communication with these influential stakeholders is vital due to their potential on future coastal development.

**4. Discussion**

The presented results encapsulate the perceptions and interpretations of survey and workshop participants regarding questions and discussions on sea level rise (SLR) within three pivotal themes across European sea basins: SLR information, hazards,

impacts, and adaptation. The varying percentage of participation among different participant profiles in each basin may contribute to the spatial differences observed in responses. However, considering the number of completed surveys, workshops





attendance, and the interactive dynamics established during these events, the results are considered to provide a representative insight into the topics investigated across European basins. It is however essential to note that, from a quantitative perspective, the participation of stakeholders and, in particular, government representatives from the Arctic Seas and Black Sea basins were notably lower than other regions, reducing the significance of the findings in these areas.


While the distinctive characteristics of each basin affect specific elements there, some shared issues highlight their importance in understanding sea level requirements for the key themes under discussion.

During almost all scoping workshops, there was a common consensus regarding the importance of *local sea level data* to accurately assess spatial sea level variations within basins, especially concerning extreme water levels. In addition to expanding existing tidal networks, it was suggested to encourage sea level monitoring through citizen science sensors. This approach not only has the potential to raise awareness among coastal communities about (extreme) sea level conditions but also leads to a more extensive and high-resolution network of coastal sea level data, addressing spatial variability effectively (e.g. Spicer et al., 2021). In addition to incorporating new data, it was acknowledged that there is an urgent need for harmonization among existing data portals providing tide gauge information, such as the Global Sea Level Observing System (GLOSS) and European data portals (e.g. Pérez Gómez et al., 2022). This also includes updating metadata related to tidal gauges, which is indispensable for accurately reconstructing and interpretating observed sea level, for instance related to vertical land movement (e.g. Latapy et al., 2023).



*Uncertainty* emerged as a recurring theme in both survey and workshops, independent of the respondent's basin of origin. Striking the right balance between effectively conveying uncertainty while providing specific data crucial for policy compliance remains a significant challenge. In this regard, Kopp et al. (2023) identify the communication of uncertainty and ambiguity as a key challenge in translating sea-level science to inform long-term coastal planning. During the workshops, some stakeholders acknowledged that an excessive emphasis on uncertainty could lead to delays or hinder progress in planning or implementation of adaptation measures. However, it is essential to recognize that the tolerance for uncertainty varies based on its intended use (e.g. long/short term applications) and the risk perceptions of individuals and groups. There tends to be a higher tolerance for uncertainty when the potential value at risk is relatively low (e.g. Hinkel et al., 2019).



In connection with this prevailing uncertainty, respondents also emphasized the importance of investigating *low-probability, high-impact SLR scenarios*. While these scenarios may be unlikely to materialize, they hold significance from a risk-management standpoint (e.g. Hinkel et al., 2015). Research-sector stakeholders underscored the need for advancing our understanding of the contributions of *ice sheets* to future SLR (e.g. Bamber et al., 2022; Van De Wal et al., 2022). Management professionals emphasize the need for regional projections that facilitate impact analysis (e.g. Dayan et al., 2021). One highlighted concern pertains to the necessity for enhanced information and data to improve current and future *regional and local sea-level change* estimations. Specifically, they emphasized the importance of assessing the local impact of *vertical land movements* on relative sea level rise. This assessment should encompass both natural and human-induced factors to accurately gauge relative SLR and, in turn, enhance assessments of SLR-induced hazards (e.g. Nicholls et al., 2021).


Participants recognized the importance of integrating comprehensive *multidisciplinary data* for assessing risks, including both exposure and vulnerability characteristics in susceptible areas, particularly in the Low Elevation Coastal Zone (LECZ). In many instances, these factors significantly influence the estimated risk (e.g. Neumann et al., 2015).


In terms of hazards and their impacts, scoping workshops consistently highlighted the need for multi-hazard risk assessments. Specifically, the workshops brought attention to *compound coastal floods*, in which elevated sea levels coincide with high river flow or heavy rainfall events. This was also identified as an impact to be considered in the open-ended questions of the survey (Fig 7). From a risk management perspective, the significance of such occurrences lies in their potential to amplify the






impact of the individual hazards and/or accumulate them within a specific region (Zscheischler et al., 2020). Within the context of this scoping process, it is crucial to recognize that SLR may influence the likelihood of occurrence and intensity of these events through anticipated changes in local extreme sea levels (e.g. Moftakhari et al., 2017), which may also affect the spatial

distribution of high-risk locations (see e.g. Bevacqua et al., 2019).

To enhance the assessment of the primary SLR-induced hazard identified by stakeholders in the global survey (Fig. 5), long-term coastal erosion, there was an emphasis on considering additional factors influencing the *sediment budget* such as sediment supplies from rivers, where the impact of river damming plays a significant role in modulating the expected erosion, especially in deltas (e.g. Ericson et al., 2006).

It is interesting to note that, while *seawater intrusion* received one of the lowest overall relevance scores in the survey (Fig 5), it was consistently brought up by participants in all scoping workshops. This emphasis is justifiable when we consider that coastal aquifers serve as critical freshwater sources for many coastal areas, and these resources face threats from both groundwater extraction and rising sea levels (e.g. Ferguson & Gleeson, 2012). The growing concern regarding SLR and its impact on seawater intrusion is evident in the recent metanalysis of seawater intrusion research by Cao et al. (2021), which

identified the impact of SLR as the most widely discussed topic. In this regard, Ketabchi et al. (2016) identified key knowledge gaps on the impacts of SLR on seawater intrusion, and recommended main aspects for future research. The relevance of this impact also aligns with the findings from open-ended survey questions, where participants highlighted water management and agriculture issues (Fig. 7).

On adaptation topics the survey responses showed slight differences in responses across basins, albeit within a relatively

narrow range (Fig 6). This variability aligns with findings from McEvoy et al. (2021), who observed regional differences in adaption planning in their analysis of European countries' approaches to sea level rise planning. One key aspect was the necessity of *tailoring sea level rise information* to different application domains, involving different stakeholders, institutions and their specific information needs (see also e.g. Hinkel et al., 2019; Durand et al., 2022). Additionally, there was a consensus on the importance of the effective communication of this information to stakeholders and the enhancement of *visualization*

*techniques* to engage local communities (e.g. Calil et al., 2021).

When *comparing responses from government and research participants* in the survey, both groups generally exhibited similar behaviour in responding to various questions. However, a significant divergence emerged regarding their views on two practice-oriented issues: the availability of impact assessments and the effectiveness of adaptation plans. Government respondents tended to be more optimistic than their research counterparts, expressing greater confidence in the availability of

high-quality and up-to-date impact assessments as well as in the effectiveness of adaptation plans. An exception to this was found in the North Sea basin, where government respondents were less confident on the flexibility of adaptation strategies than researchers. Finally, it has to be considered that while the availability of impact assessments is a quantifiable matter, the effectiveness of the adaptation plans is arguably a matter of perception for most part. In practice, the true effectiveness of these plans remains unverified until they are implemented and operational under the projected scenarios.


**Conclusions**

The combination of survey and regional workshops has effectively revealed shared knowledge gaps and needs concerning SLR across European sea basins. This assessment spans both scientific and governmental perspectives, classified into three main SLR-related themes: information on SLR, its impacts, and adaptation policies and decision making.



In terms of SLR information, notable gaps involve regional SLR projections and uncertainties, particularly related to long-term SLR induced by large-scale ice-sheet melting. Scientists view these gaps as objectives, seeking to refine regional projections and reduce uncertainty. In contrast, government users see these gaps as barriers to achieve their specific goals, for which they need accurate SLR projections for their regions and advise on how to deal with uncertainty.

Concerning hazards and impacts, shoreline erosion emerged as a prominent issue across sea basins (except in the Arctic), with
emerging issues like saltwater intrusion being recognized as undervalued and necessitating additional attention due to potential impacts on agriculture, freshwater resources and coastal ecosystems. Among these emerging issues, the role of SLR in compounding risks events, such as those related to extreme water levels and river flow, was underscored. Participants also emphasized the necessity for high-quality and updated impact assessments to inform adaptation planning to SLR.

Concerns were raised about existing adaptation plans, revealing a common perception of inefficient and inflexible strategies
to address SLR impacts. Some gaps were identified, particularly related to undervalued impacts, with urban planning being a prominent aspect needing attention. Furthermore, participants, particularly end-users, expressed the need for enhanced information dissemination and more effective communication of relevant data and information to support decision-making.

### Acknowledgements

We sincerely thank all the participants in the survey and scoping workshops for contributing their knowledge, experience and
information to this work. We thank Bernd Brüge who actively collaborated in the scoping work and comanaged the Baltic Sea workshop and Kevin Parnell who helped with the realisation of that workshop. We thank Robyn Gwee, Nathalie Dees, Geerit Hendriksen and Marta Marcos for their contribution to data included in Table 1. The work of the first author was supported by the *C3RiskMed* (PID2020-113638RB-C21) research project, funded by the Spanish Ministry of Science and Innovation (AEI/10.13039/501100011033) and by the Spanish Research Agency (AEI).


### Author contributions

JAJ wrote the paper with text contributions from GG, MM, LGP, NVR, and GW. The survey was designed, conducted and analysed by JAJ. The scoping workshops were conducted by AB, GW and BvH (North Sea and Arctic Seas); JAJ, GG and NP (Mediterranean Sea and Black Sea), MM (Baltic Sea), NVR (Eastern Atlantic). LP and MD provided support to all workshops.
All authors participated in the iterations and revisions of the paper.

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
