# Peer review of "Sea Level Rise in Europe: knowledge gaps identified through a participatory approach"

_State of the Planet, 2023_

## Author Comment (AC4)

**Reviewer #4**

The manuscript by Jiménez et al. presents an insightful and methodologically robust scoping initiative, exploring significant voids in knowledge from both government and academic viewpoints on available information on sea-level rise and its potential impacts. The analysis of stakeholder engagement processes is both valuable and compelling, highlighting the study's strengths. However, there are some areas where clarity and presentation could be improved to enhance the manuscript's overall impact and readability.

**Response to reviewer #4**

Thank you very much for the time dedicated to review the manuscript and for comments and suggestions provided.

In what follows these comments and suggestions are addressed, where each comment is presented as given (in black) and then we specify how it will be addressed (in blue).

**Specific Comments:**

**Abstract (Line 31):** The term "information gap" is somewhat ambiguous. It would be beneficial to clarify the study's primary objective more distinctly. Is it to identify gaps in climate services for non-academic fields, to pinpoint areas needing further research from scientists, or both?

Thank for your observation. In the context of our study, the two examples you provided are relevant, as we aim to identify existing gaps for both scientists and non-academic (governmental) stakeholders. We will revisit the abstract to ensure clarity and avoid introducing ambiguity.

**Link Functionality (Line 85):** The provided link to the EU Survey platform is not operational.

We have checked and, now, the link is working properly.

**Stakeholder Characteristics (Lines 101 to 106):** Introducing specific characteristics of the stakeholders earlier in the section, preferably by Line 84, would greatly benefit the narrative flow and prevent the current belated introduction of this crucial information.

We will move the description of stakeholder characteristics to an earlier point in the section, as suggested.

**Stakeholder Bias in Survey Data:** The manuscript exhibits a noticeable bias towards researcher, which skews the study's findings. Clarifying the study's main goal—whether it is identifying research gaps, improving climate services, or both—would help in understanding the significance of this bias.

Thank you for your observation. We did not intend to show bias towards researchers. Any perceived bias may have stemmed from the composition of our participant pool.

Scientists accounted for the 64 % of the total participants, which naturally led to higher representation in the responses. However, we have taken measures to address this potential bias by presenting the responses of each participant group separately in instances where they significantly differ (see e.g. Fig 3a, Fig 4, Table 3, Fig 5b, Fig 7, Table 4). Despite these efforts, we will review the text to eliminate any inadvertent bias and, if necessary, provide clearer differentiation among the various participant profiles.

**Scoping Workshop** (line 127):  It is unclear whether the workshop participants include only government representatives or also scientists – as 'stakeholders' (in line 128) could be both. If both groups are included in the workshops, a primary concern arises in the analysis of workshop outcomes, specifically regarding the clarity in distinguishing between government and researcher perspectives, as it is crucial to understand these distinct viewpoints to address the identified knowledge gaps. Providing a more detailed account of the stakeholders involved, including selection criteria and their roles, would improve readers' comprehension of the study's foundation. Or a dedicated section comparing and contrasting government and researcher perspectives could offer nuanced insights into the stakeholder engagement process.

The workshops included participants from both government representatives and scientists. The outcomes of these workshops reflect an integrated perspective, as they were designed not to solicit individual responses to specific questions but rather to serve as collaborative platforms where all participants, regardless of their background, contributed to the discussion of relevant issues. As a result, it is not feasible to discern the origin of contributions, as was possible with the survey responses. We will review the text to ensure that this perspective is well described.

**Improve Figures 2 and 3:** For Figure 2, adding a legend and title would improve its interpretability. For Figure 3, replacing numerical labels with terms like 'agree' on the Likert-scale and adding a descriptive title (the question) and a legend could make the data more accessible.

Figures will be modified to improve interpretability as suggested.

**Visual Representation of Knowledge Gaps (Section 3.2):** Presenting the main knowledge gaps/needs through visual means, such as tables or graphs, would offer a clearer, more immediate comparison across different basins.

Following the suggestion of the reviewer, we will incorporate visual aids (still to be defined if a table or a figure) to facilitate comparisons of the information presented across basins.

The manuscript provides an essential contribution to understanding stakeholder perspectives on information of sea-level rise and its impacts in Europe. By addressing the above suggestions, I believe the clarity and impact of your study can be enhanced. I look forward to seeing the revised manuscript and hope the feedback is useful.

Thank you for your positive comment.

---

## Author Comment (AC5)

**Editor**

Dear authors,

Your manuscript has received four reviews, all from solicited referees, whom I herewith would like to thank warmly for contributing their time and expertise to this first Assessment Report of the European Knowledge Hub on Sea Level Rise.

The reviews are very appreciative of the manuscript and request only minor revisions to make the manuscript acceptable. All reviews list some specific points. I would ask you to address all of them in an author comment, in addition to revising the manuscript accordingly. In particular, please consider carefully the suggestions of Referee Comment 4 on increasing clarity of the figures and think of a way to illustrate the workshop outcomes.

In addition, I went through your manuscript with an editor's perspective and am requesting some additional revisions of different kinds. I ask you to consider those and also respond to the substantial ones (i.e. not the language or typo fixes) in an author comment:

**Response to Editor**

Thank you very much for the time dedicated to edit the manuscript and for very detailed comments and suggestions provided.

In what follows these comments and suggestions are addressed, where each comment is presented as given (in black) and then we specify how it will be addressed (in blue).

This manuscript describes the overall approach as four successive steps: survey, workshops, conference, consultations. The manuscript describes the survey and workshops in detail, but not the remaining two steps. I am aware that the conference will be features in an introduction section to the report. A cross referencing (for now just with a placeholder) should be put in. Also, it would be good to describe, or at least indicate, how the other two components contributed to the co-design, and explain why these components are not described in this manuscript. Alternatively, add a small sub-chapter for this to have the entire co-design process described in one place.

The Editor accurately outlines the steps involved in the scoping process. Our focus was primarily on two of these steps, survey and workshops. We assumed that these additional processes would be detailed in an introductory section. However, if providing a description of the additional steps would enhance understanding of the adopted framework, we are open to incorporating a brief overview of the overall processes, along with details on the two supplementary steps.

Chapter 3.1 on the survey results provides percent numbers from aggregated survey results, e.g. in line 152ff "32% of the respondents expressed that a substantial portion of this information is missing ...". These numbers cannot be retraced anywhere. Fig. 3 is referenced at the end of this first paragraph, but the numbers provided can actually not be seen there, which is confusing. Can this be made more consistent? Reference is also made to Table S2 (actually presumably meaning S1) but here I struggled to follow how the %-numbers were generated. I feel that the manuscript should somehow provide access to the raw survey data. This would also adhere to good practice of transparency and data accessibility.

In the text we referenced the responses in terms of % for both types of participants, whereas the figure shows the corresponding % values for all participants integrated (shown at the bottom figures in colored scale indicating %). The idea behind this was to provide complementary information in both the text and figures.

The editor's comment regarding the reference to Table S1 is correct. It will be changed.

Regarding access to the raw survey data, we will include a properly anonymized copy of data in the supplementary material.

The Knowledge Hub on Sea Level Rise is usually abbreviated without a hyphen, just "KH SLR". For consistency, I suggest to adopt this for your paper also.

We will do it.

As this is about the European Knowledge Hub on Sea Level Rise, I suggest to use British spelling, i.e. the more common use of s instead of z in words like emphasise, prioritise, utilise, visualise, standardise, harmonise, materialise, ...

We will check the use of the British spelling throughout the manuscript and avoid mixing US and British styles.

Throughout the manuscript, the word "significant" is used in different ways. Given that the manuscript has the analysis of survey data as a basis, please use consistently only for statistical or empirical significance. In other contexts, other adjectives like "relevant" might be more fitting (e.g. line 96, 208, 235, and elsewhere).

We will check the wording when referring to results of the surveys to avoid confusions.

Below are some minor comments and editorial fixes:

All these editorial corrections will be adopted. In addition to this, the final version will undergo a complete grammatical revision.

TEXT:

line 24: replace "paper" with "chapter"

line 27: add "sea" to make it European sea basins

line 28: present tense "is" instead of "was"

line 30: long-term SLR: Can you give a semi-quantitative indication what you mean, like multi-decadal? Or multi-centennial? Or end of century? ...

line 37: especially end-users [add "of sea-level knowledge"]

line 41 and following: "uneven coverage in the knowledge ..." This sounds if we had sufficient information which is just a bit unevenly distributed. Is this the case? Or is applicable knowledge missing everywhere? If so, make sure to make this clear.

line 46: The word "is" is missing in "This IS  the ambition ..."

line 52: "three key components": Further below it lists four components, including consultations. Please make consistent.

line 53: fix wording to "pertinent in Europe"

line 77: Fix typo 19502 to 1950.

line 99: Fix typo "on IPCC reports" to either in or of.

line 113: Please explain the logic of the maximum score of 600.

line 129: Delete "the" in "each of the Europe's ..."

line 130: For consistency with elsewhere in the text, change ocean basins to sea basins.

line 133: Once introduced the abbreviation KH SLR you may as well use it consistently throughout the text. Same applies to SLR

line 145/6: Be clearer how workshop participation was achieved and interest handled. By invitation only? I believe that the workshops had been advertised also.

line 162: Please choose the more formal "it is" instead of "it's".

line 164: For clarity add "sea" to make it "different sea basins".

line 242: "existing high-quality and up-to-date assessments of SLR-induced impacts" is not a statement. Please fix the sentence.

lines 264 and 514 (and elsewhere): Consider replacing optimistic with positive.

line 269: replace "an" with "this"

line 270: Consider making it "distribution of most frequent responses".

line 319: Please give an indication what sea-level milestones are. It seems to be the first mention in this manuscript here.

line 323: Replace its' with 'it is".

line 347: Budgets should probably be plural?

line 393: Rephrase to "... values were considered a top priority."

line 399: Remove "Sea" to make it "non-European coastal areas".

line 411/412: As you have introduced the abbreviation SLR already, you may use it throughout the text, i.e. also in line 412. On the other hand, I suggest to avoid using abbreviations in headers (like line 411 and elsewhere).

line 448: Make it "European sea basins"

line 450: Replace in with for.

line 455: Please give a better indication what citizen science sensors are.

line 471: Also italise "low-probability".

line 495: For consistency replace seawater with saltwater.

line 540: The spell Brügge with double-g.

line 549: LP should be LGP.

FIGURES AND TABLES:

Figure 1 caption: Please be a bit more elaborate and precise what this is about (i.e. the KH's distinction of sea and ocean basins to structure the co-design consultations).

We'll do it.

Table 1: This table contains interesting information but is not used in the text. I recommend to extract a few points that point to the differences and/or characteristics of the sea basins, which is relevant for the discussion then also.

We will reference the observed differences across basins as presented in the table to provide context for some of the results.

Figure 2: Line 103ff describes the distinction of two professional groups. These could be indicated also in figure 2b to make this clearer.

Fig 2 b illustrates the members of each basic type of stakeholders. We will slightly modify it by incorporating two colors to indicate which stakeholders contribute to each of the two main groups used throughout the manuscript, government and scientists.

Table 2: heading "N registered attendees" could be just "attendees" or at least just "registered attendees".

We'll do it.

Supplement Tables: Numbering is inconsistent between text and supplement. It seems that one Table has been removed in the process but numbers in the text have not been corrected.

We will check the numbering.

Figure 3: panel (a) could get a heading such as 'Information deficit' to make it clearer what topic this is about.

Table 3: caption: Can it be made more explanatory, e.g. by starting with "Rankings of perceived knowledge gaps based on ..."?

The heading and caption will be checked and revised to improve the description.

Figure 5: In analogy to Fig. 3, can the Likert-chart be put first (i.e. chart a here) and the colour matrix follow as chart b?

Figure 5: Can the Linkert graph get a keyword such as "Impact information availability"?

Figure 5: Black and Arctic Seas have no government values? Why are they then not identical with the grey bar mean?

As mentioned in the text, the number of government participants from these two basins was too low to be significative. We will also add a note in the figure on this.

Figure 5: caption: For clarity in relation to the other chart in this figure, could you add "specific" to make it read "Relevance of specific SLR-induced impacts ..."

Figure 5: caption: Please replace "colour bar" with "grey bar".

We will review Figure 5 taking into account Editor's suggestions.

Figure 6: Consider adding keywords to the chrats for quicker grasping of the content, like effectiveness, flexibility, and NBS appropriateness.

We'll do it.

Figure 7: caption: I can't find the question "Are there other decisions/purposes for which you currently don't consider SLR, but for which you think it would be important to do so?" in the supplement.

Thanks. We will add it.

SUPPLEMENT:

Please structure the supplement more clearly to make it easier to find the relevant information. I suggest to use the numbering 1.-8. on the cover page also within the summplement. Also, please number all elements in the supplement document.

We'll do it.

Figure S1 ICES ecoregions should probably bot be S1 any more.

The table below that ICES ecoregions seems partly wrong. Several of the Atlantic labels should probably be Arctic. I would also suggest that you use the exactly same nomenclature as in Figure 1, i.e. if you said East Atlantic Seas there, also name it that way in the supplement.

We will check the equivalence of both regionalization and correct tables to avoid confusion.

I look forward to receiving your revised manuscript.

Kind regards,

Thorsten Kiefer

---

## Author Response (AR1)

**Reviewer # 1**

This work greatly complements existing sporadic knowledge of the state of Europe's adaptation to SLR. By combining both a basin-by-basin approach and a pan-European one, it provides an excellent assessment of trends in the application of existing knowledge, as well as gaps in actionable knowledge. This paper is also remarkably comprehensive and concise.

**Response to reviewer #1**

Thank you very much for the time dedicated to review the manuscript and for comments and suggestions provided.

In what follows these comments and suggestions are addressed, where each comment is presented as given (in black) and then we specify how it will be addressed (in blue).

To further improve the rendering of these findings, it would be useful to report on the results obtained (if any) concerning the consideration given by stakeholders to the feasibility and effectiveness of the range of adaptation responses/solutions in the definition of strategies (not just the NBS typology, for which this analysis is already provided).

This point is significant as it could offer additional insights into stakeholders' perceptions regarding various adaptation options. However, it falls outside the intended scope of the survey, and thus, no specific mention was made of other options. The explicit reference to NBS was made to assess how the prevailing approach to adaptation measures, both within the EU and globally, is influencing stakeholders' perspectives. In any case, it is worth noting that the most selected response was that NBS are appropriate in selected cases, recognizing the need to consider different adaptation strategies.

Another point that deserves clarification: only the analysis of the Mediterranean region mentions social considerations. It would be interesting to elaborate on this point. Did respondents and participants from other regions not mention it? Are there underlying reasons for this Mediterranean specificity?

The mention of social aspects occurred in the basin-specific workshops. Consequently, the reference to the different topics reflects the interest and concerns of the diverse participants involved. Regarding the specific case highlighted by the reviewer, social aspects were indeed discussed in relation to both impacts (*the accurate estimation of the vulnerability of the densely populated coastal zones, and their exposure and values were considered as a top priority*) and adaptation (*Any adaptation policy should take into account social factors and community engagement, ensuring a participatory decision-making process where diverse stakeholders have a voice*). It is important to note that while these aspects were explicitly mentioned in the Mediterranean workshop, they remain relevant across all European basins (and worldwide). Their applicability and significance are widely accepted, though the explicit mention may vary among workshops.

To provide further context, we have included a comment acknowledging the applicability of these observations across all European sea basins. Thus, the following sentence was added to the Discussion section when summarizing the results related to adaptation:

"The relevance of taking *social factors* into account when formulating adaptation strategies was also noted, since barriers and limits to adaptation often stem from social aspects rather than purely technical factors (e.g. Adger et al. 2009; Hinkel et al. 2018; Galluccio et al. 2024)".

**Reviewer # 2**

The article gives very interesting and useful information on the different situations in the European countries on information availability and decission making in relation to sea level rise. Good and extensive overview of the gathered information.

**Response to reviewer #2**

Thank you very much for the time dedicated to review the manuscript and for comments and suggestions provided.

In what follows these comments and suggestions are addressed, where each comment is presented as given (in black) and then we specify how it will be addressed (in blue).

The importance of exchange information on data, projections and plans could be worked out a bit more. If neighboring countries come to different conclusions of the rate of slr and the urgency this could lead to misunderstanding and confusion in society. If countries have plans to adapt to a rising sea level that effect and/or do not match with the plans of a neighbouring country, this could have severe adverse consequences.

The reviewer's observation is indeed sound. Discrepancies on these matters may lead to variations in the perceived urgency and necessity for a unified response to a global issue demanding action at a transnational level. We will incorporate a cautionary comment regarding this aspect in the discussion section. However, it is important to note that the manuscript is reflecting the current perception of the stakeholders, while companion papers are analyzing in depth the current state of data/information availability on SLR and adaptation policies across Europe. We will also make reference to those papers.

The data suggest a comforting similar sense of urgency and prioritization of sea level rise issues on slr, specifically on uncertainty. Not mentioned is the difference in time horizon that there might be for planning and decision making in relation to the scientific scope. For policy makers there is a need for systems that helps Risk informed decision making. This could be a joint interest to work on.

Once more, the reviewer's observation is sound. While one of the primary focus of concerns was on aspects such as uncertainty, it is essential to note that the aspects explicitly mentioned are not exhaustive. In fact, any addressed in this manuscript must be contextualized and supplemented by its status across European basins, as extensively analyzed in companion papers focusing on SLR information, impacts, adaptation and governance. As mentioned before, we will include a comment making reference to those papers to contextualize the perception of stakeholders again the current status of knowledge in Europe.

To address the two comments above, the following text was included as a concluding paragraph in the Discussion section:

"Lastly, it is important to acknowledge that the results presented herein represent the prevailing perceptions of stakeholders across European sea basins regarding various aspects to SLR. These findings should be interpreted with the other chapters of this report, where detailed analyses are provided on the current state of data/information availability on SLR (Melet et al. 2024), the resulting impacts (van de Wal et al. 2024), adaptation policies (Galluccio et al. 2024) and the governance landscape (Bisaro et al. 2024) throughout Europe sea basins".

The article is providing a good basis for further cooperation and is hopefully providing for next steps. On exchanging information on monitoring, methods for projections, instruments, approach, process of preparing adaption plans, decisions making, etc.. It would give the article more value if suggestions would be added for further steps or cooperation.

Thank you for the comment. Indeed, during some of the workshops, participants highlighted the importance of international cooperation to share experiences and knowledge. Furthermore, the variations observed in responses across different regions as well as common concerns and interests suggest the value of such knowledge transfer to enhance harmonization across European basins. We will include a recommendation in this sense as concluding remark.

The following text was included as a final conclusion:

"Finally, stakeholders underscored the significance of international collaboration for sharing experiences and expertise on the different aspects related to sea level rise. Moreover, the observed discrepancies in responses among different regions, alongside shared concerns and interests, underscore the importance of knowledge exchange to foster harmonization across European sea basins".

Some small details:

Fig 2b: Other (what is added after 'other' can be skipped; the explanations for other is not mentioned below)

Thanks. It will be removed or changed to "other locations". It was included to indicate that some respondents were from other basins outside Europe.

Fig 6: result presentations are likely to give a reverse idea while no plans or no ineffective plans are figured green, while red stands for good, sufficient and effective plans.

Thanks for the comment. While it's understandable that our perception of colors often associates green with positive and red with negative, it is important to note that in this context, colors merely represent percentages. We have employed a consistent scale

(included in the figure) throughout the manuscript to minimize any potential confusion. Nevertheless, we will assess the feasibility of adjusting the scale to minimize such perceived confusion.

**Reviewer # 3**

To my opinion, this work contributes to a large extent to an important topic of identifying critical gaps in available information on regional SLR and its potential impacts across European basins and provides a comprehensive analysis of knowledge requirements and areas necessitating further research. Based on relevant survey and workshops, it emphasizes the role of participatory approach and turns attention to regional disparities and lack of data, cooperation and access to information on SLR at all levels.

This paper may be of interest to policy makers, coastal planners, and stakeholders at large, and it definitely contributes to the ongoing process of harmonization of efforts in terms of policy and data collection between European regional seas (under MSFD-related directives, under UNEP Regional Seas Program, bilateral arrangements between regional sea conventions etc.) and setting a scene for further steps to be implemented for improving local and regional SLR projections, as well as improvements in physical science and data, hazards and impacts, adaptation policies and decision-making. The information analyzed in the article shows increasing recognition of nature-based solutions (NBS) and role of ecosystem management in addressing the SLR, which can be further used by regional policy-makers.

**Response to reviewer #3**

Thank you very much for the time dedicated to review the manuscript and for comments provided.

In what follows these comments are addressed, where the comment is presented as given (in black) and then we specify how it will be addressed (in blue).

İt could be still good to see a vision of potential instruments and approaches of regional cooperation on SLR in order to deal with challenges and gaps described in this article. Overall, this work is very comprehensive and eye-opening.

The reviewer is right and, in fact, this comment is related to one raised by reviewer#2 about the need for further cooperation on the issue. As we mention there, during some of the workshops, participants highlighted the importance of international cooperation to share experiences and knowledge. Furthermore, the variations observed in responses across different regions as well as common concerns and interests suggest the value of such knowledge transfer to enhance harmonization across European basins. We have included a recommendation in this sense as concluding remark.

**Reviewer #4**

The manuscript by Jiménez et al. presents an insightful and methodologically robust scoping initiative, exploring significant voids in knowledge from both government and academic viewpoints on available information on sea-level rise and its potential impacts. The analysis of stakeholder engagement processes is both valuable and compelling, highlighting the study's strengths. However, there are some areas where clarity and presentation could be improved to enhance the manuscript's overall impact and readability.

**Response to reviewer #4**

Thank you very much for the time dedicated to review the manuscript and for comments and suggestions provided.

In what follows these comments and suggestions are addressed, where each comment is presented as given (in black) and then we specify how it will be addressed (in blue).

**Specific Comments:**

**Abstract (Line 31):** The term "information gap" is somewhat ambiguous. It would be beneficial to clarify the study's primary objective more distinctly. Is it to identify gaps in climate services for non-academic fields, to pinpoint areas needing further research from scientists, or both?

Thank for your observation. In the context of our study, the two examples you provided are relevant, as we aim to identify existing gaps for both scientists and non-academic (governmental) stakeholders. We have included the following sentence to ensure clarity of the scope.

"It considers viewpoints from both scientific and policy perspectives, engaging stakeholders from academia/research and government sectors."

**Link Functionality (Line 85):** The provided link to the EU Survey platform is not operational.

We have checked and, now, the link is working properly.

**Stakeholder Characteristics (Lines 101 to 106):** Introducing specific characteristics of the stakeholders earlier in the section, preferably by Line 84, would greatly benefit the narrative flow and prevent the current belated introduction of this crucial information.

The description of stakeholder characteristics has been moved to an earlier point in the section, as suggested.

**Stakeholder Bias in Survey Data:** The manuscript exhibits a noticeable bias towards researcher, which skews the study's findings. Clarifying the study's main goal—whether it is identifying research gaps, improving climate services, or both—would help in understanding the significance of this bias.

Thank you for your observation. We did not intend to show bias towards researchers. Any perceived bias may have stemmed from the composition of our participant pool. Scientists accounted for the 64 % of the total participants, which naturally led to higher representation in the responses. However, we have taken measures to address this potential bias by presenting the responses of each participant group separately in instances where they significantly differ (see e.g. Fig 3a, Fig 4, Fig 5a, Fig 7, Table 3, Table 4). We have revised the text to avoid any unintentional bias and, when necessary and possible, to reflect a clearer differentiation between the different participant profiles.

**Scoping Workshop** (line 127):  It is unclear whether the workshop participants include only government representatives or also scientists – as 'stakeholders' (in line 128) could be both. If both groups are included in the workshops, a primary concern arises in the analysis of workshop outcomes, specifically regarding the clarity in distinguishing between government and researcher perspectives, as it is crucial to understand these distinct viewpoints to address the identified knowledge gaps. Providing a more detailed account of the stakeholders involved, including selection criteria and their roles, would improve readers' comprehension of the study's foundation. Or a dedicated section comparing and contrasting government and researcher perspectives could offer nuanced insights into the stakeholder engagement process.

The workshops included participants from both government representatives and scientists. The outcomes of these workshops reflect an integrated perspective, as they were designed not to solicit individual responses to specific questions but rather to serve as collaborative platforms where all participants, regardless of their background, contributed to the discussion of relevant issues. As a result, it is not feasible to discern the origin of contributions, as was possible with the survey responses. The text has been reviewed to ensure that it reflects both perspectives.

**Improve Figures 2 and 3:** For Figure 2, adding a legend and title would improve its interpretability. For Figure 3, replacing numerical labels with terms like 'agree' on the Likert-scale and adding a descriptive title (the question) and a legend could make the data more accessible.

Figure 2a & 2b have a detailed legend describing their content.

Figure 3a. We have added a qualitative description of numbers in the scale following the suggestion of the reviewer (this has also been applied to Fig 5a). The legend includes the question.

**Visual Representation of Knowledge Gaps (Section 3.2):** Presenting the main knowledge gaps/needs through visual means, such as tables or graphs, would offer a clearer, more immediate comparison across different basins.

While the reviewer's suggestion is sound, the key elements identified from the surveys do not allow a clearer distinction between basins, as discussions often centered around common topics. Consequently, graphical visualization may not offer additional

value for comparison beyond what is provided in the text. However, to aid in identifying the main topics highlighted in each basin, we have italicized key words.

The manuscript provides an essential contribution to understanding stakeholder perspectives on information of sea-level rise and its impacts in Europe. By addressing the above suggestions, I believe the clarity and impact of your study can be enhanced. I look forward to seeing the revised manuscript and hope the feedback is useful.

Thank you for your comments.

**Editor**

Dear authors,

Your manuscript has received four reviews, all from solicited referees, whom I herewith would like to thank warmly for contributing their time and expertise to this first Assessment Report of the European Knowledge Hub on Sea Level Rise.

The reviews are very appreciative of the manuscript and request only minor revisions to make the manuscript acceptable. All reviews list some specific points. I would ask you to address all of them in an author comment, in addition to revising the manuscript accordingly. In particular, please consider carefully the suggestions of Referee Comment 4 on increasing clarity of the figures and think of a way to illustrate the workshop outcomes.

In addition, I went through your manuscript with an editor's perspective and am requesting some additional revisions of different kinds. I ask you to consider those and also respond to the substantial ones (i.e. not the language or typo fixes) in an author comment:

**Response to Editor**

Thank you very much for the time dedicated to edit the manuscript and for very detailed comments and suggestions provided.

In what follows these comments and suggestions are addressed, where each comment is presented as given (in black) and then we specify how it will be addressed (in blue).

This manuscript describes the overall approach as four successive steps: survey, workshops, conference, consultations. The manuscript describes the survey and workshops in detail, but not the remaining two steps. I am aware that the conference will be features in an introduction section to the report. A cross referencing (for now just with a placeholder) should be put in. Also, it would be good to describe, or at least indicate, how the other two components contributed to the co-design, and explain why these components are not described in this manuscript. Alternatively, add a small sub-chapter for this to have the entire co-design process described in one place.

The Editor accurately outlines the steps involved in the scoping process. Our focus was primarily on two of these steps, survey and workshops. We assumed that these additional processes would be detailed in an introductory section.

We have included the following text in the Introduction:

"The Sea Level Conference 2022, promoted by the KH SLR, focused on evaluating and exchanging scientific knowledge and policy development regarding SLR in European

coastal regions. Rooted in findings from the survey and scoping workshops, it featured insights from experts from the Knowledge Hub as well as invited experts and policymakers from each basin, through a combination of keynotes, panels, and posters. The outcomes aimed to provide accessible and updated knowledge tailored to users across European basins, addressing the needs of policymakers, coastal planners, and stakeholders. While this chapter focusses on the first two components, further description on the other components can be found in the introduction of this report (Reference to Introductory chapter, 2024)".

Chapter 3.1 on the survey results provides percent numbers from aggregated survey results, e.g. in line 152ff "32% of the respondents expressed that a substantial portion of this information is missing ...". These numbers cannot be retraced anywhere. Fig. 3 is referenced at the end of this first paragraph, but the numbers provided can actually not be seen there, which is confusing. Can this be made more consistent? Reference is also made to Table S2 (actually presumably meaning S1) but here I struggled to follow how the %-numbers were generated. I feel that the manuscript should somehow provide access to the raw survey data. This would also adhere to good practice of transparency and data accessibility.

In the text we referenced the responses in terms of % for both types of participants, whereas the figure shows the corresponding % values for all participants integrated (shown at the bottom figures in colored scale indicating %). The idea behind this was to provide complementary information in both the text and figures.

The editor's comment regarding the reference to Table S1 is correct. It has been changed.

Regarding access to the data, the following "Data availability statement" has been added.

"The collected data are not publicly available as the participants of this study did not give written consent for their data to be shared publicly. Anonymized data can be provided by the corresponding author upon request".

The Knowledge Hub on Sea Level Rise is usually abbreviated without a hyphen, just "KH SLR". For consistency, I suggest to adopt this for your paper also.

Done.

As this is about the European Knowledge Hub on Sea Level Rise, I suggest to use British spelling, i.e. the more common use of s instead of z in words like emphasise, prioritise, utilise, visualise, standardise, harmonise, materialise, ...

Checked.

Throughout the manuscript, the word "significant" is used in different ways. Given that the manuscript has the analysis of survey data as a basis, please use consistently only for statistical or empirical significance. In other contexts, other adjectives like "relevant" might be more fitting (e.g. line 96, 208, 235, and elsewhere).

Done.

Below are some minor comments and editorial fixes:

All the recommended editorial corrections have been adopted.

TEXT:

line 24: replace "paper" with "chapter" Done.

line 27: add "sea" to make it European sea basins Done.

line 28: present tense "is" instead of "was" Done.

line 30: long-term SLR: Can you give a semi-quantitative indication what you mean, like multi-decadal? Or multi-centennial? Or end of century?

Now it reads "… particularly related to long-term (from multi-decadal to end of century) SLR …"

line 37: especially end-users [add "of sea-level knowledge"] Done.

line 41 and following: "uneven coverage in the knowledge ..." This sounds if we had sufficient information which is just a bit unevenly distributed. Is this the case? Or is applicable knowledge missing everywhere? If so, make sure to make this clear.

We have changed to "…there remains an uneven distribution in both the knowledge and application of sea level science,…"

line 46: The word "is" is missing in "This IS  the ambition ..." Done.

line 52: "three key components": Further below it lists four components, including consultations. Please make consistent. Done.

line 53: fix wording to "pertinent in Europe" Done.

line 77: Fix typo 19502 to 1950. Done.

line 99: Fix typo "on IPCC reports" to either in or of. Done.

line 113: Please explain the logic of the maximum score of 600. Done.

line 129: Delete "the" in "each of the Europe's ..." Done.

line 130: For consistency with elsewhere in the text, change ocean basins to sea basins. Done.

line 133: Once introduced the abbreviation KH SLR you may as well use it consistently throughout the text. Same applies to SLR Done.

line 145/6: Be clearer how workshop participation was achieved and interest handled. By invitation only? I believe that the workshops had been advertised also. Now it reads "Participants ranged from stakeholders from each European sea basin who

participated in the survey, to others who responded to both personalized and public invitations".

line 162: Please choose the more formal "it is" instead of "it's". Done.

line 164: For clarity add "sea" to make it "different sea basins". Done.

line 242: "existing high-quality and up-to-date assessments of SLR-induced impacts" is not a statement. Please fix the sentence. Done.

lines 264 and 514 (and elsewhere): Consider replacing optimistic with positive. Done.

line 269: replace "an" with "this" Done.

line 270: Consider making it "distribution of most frequent responses". Done.

line 319: Please give an indication what sea-level milestones are. It seems to be the first mention in this manuscript here.

Changed to "associated sea-level projections".

line 323: Replace its' with 'it is". Done.

line 347: Budgets should probably be plural? Done.

line 393: Rephrase to "... values were considered a top priority." Done.

line 399: Remove "Sea" to make it "non-European coastal areas". Done.

line 411/412: As you have introduced the abbreviation SLR already, you may use it throughout the text, i.e. also in line 412. On the other hand, I suggest to avoid using abbreviations in headers (like line 411 and elsewhere). Done.

line 448: Make it "European sea basins" Done.

line 450: Replace in with for. Done.

line 455: Please give a better indication what citizen science sensors are. We have added "… it was suggested to encourage sea level monitoring through citizen science sensors such as low-cost GNSS receivers and pressure sensors (Ahmed et al. 2023)".

line 471: Also italise "low-probability". Done.

line 495: For consistency replace seawater with saltwater. Done.

line 540: The spell Brügge with double-g. Done.

line 549: LP should be LGP. Done.

FIGURES AND TABLES:

Figure 1 caption: Please be a bit more elaborate and precise what this is about (i.e. the KH's distinction of sea and ocean basins to structure the co-design consultations).

Done.

Table 1: This table contains interesting information but is not used in the text. I recommend to extract a few points that point to the differences and/or characteristics of the sea basins, which is relevant for the discussion then also.

The aim of table 1, along with the table included in section 7 of the supplementary material, was to offer basic figures concerning European sea basins. These data serve to provide context for comparing factors such as the population residing in potentially vulnerable areas to SLR (LECZ), as well as the distribution between rural and urban coastal environments. Alongside the map illustrating their extent, this enables readers to discern the relative representativeness of corresponding results from the survey and workshops. Given that the provided information is quite basic, extracting portions of it for inclusion in the text would not offer any additional advantage beyond what is already presented in the table, It would essentially amount to redundant repetition of the same data.

Figure 2: Line 103ff describes the distinction of two professional groups. These could be indicated also in figure 2b to make this clearer.

Fig 2 b illustrates the members of each basic type of stakeholders. We have slightly modified it by incorporating two colors to indicate which stakeholders contribute to each of the two main groups used throughout the manuscript, government and scientists.

Table 2: heading "N registered attendees" could be just "attendees" or at least just "registered attendees".

Done.

Supplement Tables: Numbering is inconsistent between text and supplement. It seems that one Table has been removed in the process but numbers in the text have not been corrected.

Checked and corrected.

Figure 3: panel (a) could get a heading such as 'Information deficit' to make it clearer what topic this is about.

The figure legend explicitly mentions the question addressed.

Table 3: caption: Can it be made more explanatory, e.g. by starting with "Rankings of perceived knowledge gaps based on ..."?

The heading and caption will be checked and revised to improve the description.

Figure 5: In analogy to Fig. 3, can the Likert-chart be put first (i.e. chart a here) and the colour matrix follow as chart b?

Done.

Figure 5: Can the Linkert graph get a keyword such as "Impact information availability"?

The figure legend explicitly mentions the question addressed.

Figure 5: Black and Arctic Seas have no government values? Why are they then not identical with the grey bar mean?

As indicated in the text, the participation of government representatives from these two basins was insufficient to be significant (2 government representatives each). We have included a note in the caption regarding this. Nonetheless, given the relatively low total number of respondents, their different response with respect to research representatives account for the lack of alignment with the grey bars.

Figure 5: caption: For clarity in relation to the other chart in this figure, could you add "specific" to make it read "Relevance of specific SLR-induced impacts ..." Done.

Figure 5: caption: Please replace "colour bar" with "grey bar". Done.

Figure 6: Consider adding keywords to the chrats for quicker grasping of the content, like effectiveness, flexibility, and NBS appropriateness. Done.

Figure 7: caption: I can't find the question "Are there other decisions/purposes for which you currently don't consider SLR, but for which you think it would be important to do so?" in the supplement.

It was attached to another question. Now both are more clearly highlighted.

SUPPLEMENT:

Please structure the supplement more clearly to make it easier to find the relevant information. I suggest to use the numbering 1.-8. on the cover page also within the summplement. Also, please number all elements in the supplement document.

Done.

Figure S1 ICES ecoregions should probably bot be S1 any more.

The table below that ICES ecoregions seems partly wrong. Several of the Atlantic labels should probably be Arctic. I would also suggest that you use the exactly same nomenclature as in Figure 1, i.e. if you said East Atlantic Seas there, also name it that way in the supplement.

Checked and corrected.

I look forward to receiving your revised manuscript.

Kind regards,

Thorsten Kiefer

---

## Author Response (AR2)

Dear authors,

Thank you for responding to the referee and editor comments and revising the manuscript accordingly. I feel that the responses and revisions are overall satisfactory. I am therefore accepting the manuscript subject only to technical corrections, which I trust you address reliably without needing another round of my approval as handling editor.

A note of reassurance to the referees: As the authors argue in their response, some of the good ideas shared by referees go beyond the study design and could not be included after the surveys and meetings had already been conducted. However, the Knowledge Hub on Sea Level Rise will duly take note of these ideas for consideration in a possible second assessment cycle and report.

The technical corrections I am asking the authors to carry out are the following:

Thank you for dedicating your time to review the manuscript and for comments and suggestions provided. They have been addressed as detailed below.

Throughout: As mentioned before, please use British spelling with s instead of z given it is a European report, for words like emphasise, organisation, categorise, and many more.

Done: Checked throughout the entire manuscript.

Line 28: "academia/research". Please replace the slash (the meaning of which is ambiguous) with a word or decide for either academia or research.

Done: academia and research.

Line 68: Please replace "Member State representatives" with "member country representatives", just to avoid confusion with EU Member States and their terminology.

Done.

Line 73ff Sentence "While this chapter focusses on the first two components, ..." Unclear to what "the first two components" (namely the online survey and the sea-basin specific workshops) and "the other components" (conference and member countries consultation) this refers to. Also, please improve the logic of the end of this paragraph with the beginning of the subsequent paragraph. The original cohesion has suffered after text has been injected.

We have removed the aforementioned sentence ("While ….") and now the transition to the following paragraph is more natural.

Line 97: "In total, we received responses from 200 participants ..." Did you receive responses from exactly 200 participants? If not, please give the exact number. If not possible for some reason, indicate that responses were from approximately (or almost or more than, or ...) participants.

We received 200 responses.

Line 99: Is the first reference to Figure 2 correct or did you actually want to refer to the sea basins map of Figure 1?

It is correct, it refers to Fig 2 left.

Lines 101/102: Can you please be consistent with rounding between the two groups to close the 100%? Or is the missing 1% due to those respondents that didn't undisclosed their organisation type? If so, please indicate this to close the 100%. Also, if you give percentage values like 64%, there is no need to use "about".

Done: 94% and 6%.

Caption Figure 2: Correct from "colour" to "black" and be consistent with either singular or plural "bar/s": (a) Breakdown of respondents by sea basin (solid black bars: % government respondents; cross-hatched bars: % research respondents).

Done.

Figure 2: Here, the author response has partly rebutted the former request by myself and by referee #4 to add meaningful title keywords (and potentially legends) into the figure boxes, with the argument that the information can be found in the caption. I once again ask to consider whether more title keywords can be added into the boxes for easier readability given that the assessment report tries to reach different stakeholders for whom we wish to make the information comfortably accessible; e.g. Figure 2 box a could be labelled "(a) respondents by sea basin" box b could be "(b) respondents by organisation type". Thank you for having added such keywords in figure 6. Please also consider title keywords for figures 3 and 5, although I appreciate that it might be harder to find sufficiently succinct ones there.

Done: Keywords have been included in Figs 2, 3 and 5.

Line 155: Replace "both ... and" with "either ... or".

Done

Line 467: Spell out GNSS.

Done: Global Navigation Satellite System (GNSS)

Line 537: Write "European" or "Europe's" instead of "Europe".

Done: European

Line 569: Replace international (which has in Europe a specific meaning limited to collaboration beyond Europe) to "trans-national".

Done

Figure 6: The author response says that authors would assess the feasibility of adjusting the colour scale to minimise perceived confusion. I see no change in the figure and ask the authors to check whether this is because no better solution could be found or whether it has been forgotten to look into alternative colour scales.

As we informed the former reviewer, the colours in the figure represent percentages. We used a consistent scale (included in the figure) throughout the manuscript to minimize confusion. Changing the colour scale, whether to red or another colour (tested) would not solve the issue, as both positive and negative responses are depicted using the same colour scheme.